# Stents in Veterinary Medicine

**DOI:** 10.3390/ma16041480

**Published:** 2023-02-10

**Authors:** Szymon Graczyk, Robert Pasławski, Arkadiusz Grzeczka, Liza Litwińska, Dariusz Jagielski, Urszula Pasławska

**Affiliations:** Institute of Veterinary Medicine, Department of Biological and Veterinary Sciences, Nicolaus Copernicus University, Lwowska 1, 87-100 Toruń, Poland

**Keywords:** stent, tracheal stenting, airway collapse, ureter obstruction

## Abstract

Stenting in veterinary medicine has been a rapidly growing method of interventional surgery for several years. This procedure is usually performed in the respiratory and urinary tracts, but there are cases of stenting of blood vessels or gastrointestinal structures. It is based on maintaining the permeability of a given tubular structure, thus allowing the passage of gas or liquid. This procedure is often performed as a first-line treatment in situations where pharmacological agents do not work and as an alternative method, often cheaper than the classically performed ones. There are also cases where stenting is used as a palliative treatment, e.g., to enable defecation in colonic obstruction due to tumour infiltration of the colon wall. Stenting is often a life-saving or comfort-improving procedure for animals, but one should also be aware of possible postoperative complications and be prepared for any adversity. For this reason, this review provides an insight into the current knowledge in veterinary medicine about stenting and the consequences associated with this procedure.

## 1. Introduction

Stents are tubular ducts made of non-invasive materials designed to maintain the continuous flow of air through the airway lumen, or various types of fluid in the case of the urinary and circulatory systems. Their use in medicine has become increasingly important since 1986, when one of the first self-expandable stents was implanted in coronary arteries with long-term positive results [1]. At the same time, stenting was being attempted in dogs but with unsatisfactory results [2,3]. This encouraged researchers to explore new possibilities for the use of stents in human medicine and later in veterinary medicine. They have become an attractive alternative for several procedures, including tracheal collapse or urinary tract obstruction [4,5,6], due to the reduction in adverse effects compared to previous procedures. Stents have also found application in procedures where traditional methods have become ineffective due to the atypical morphology of the structure in question, e.g., during the occurrence of a persistent arterial duct with two different diameters at the ends of the vessel [7]. Stent placement has also found its way into palliative treatment, which has significantly alleviated the clinical symptoms of concomitant disease, i.e., in the case of stenting of a part of the colon affected by mucosal cancer, making defecation difficult [8]. The locations of the described stent sites are shown in Figure 1.

Currently, stents made of different types of materials, including various metallic, polymeric, and silicone materials, as well as biodegradable fibres, are used in veterinary medicine [9,10]. In new generations of stents, these materials are coated with various types of layers that perform additional functions. These include stents that release immunomodulatory drugs (paclitaxel, biolimus, sirolimus, arsenic trioxidate, or zotarolimus) or antibiotics (cefotaxime, triclosan), which are particularly useful in highly invasive procedures that often induce inflammation [10,11,12,13,14,15,16,17,18,19]. The most commonly used seems to be the self-expandable nitinol stent (nickel-titanium alloy with a ratio of 55:45%) with different types of plexus, as shown in Figure 2 [20]. In favour of its use is spontaneous passivation, which prevents corrosion of the material and significantly increases its biocompatibility with surrounding tissues [21]. Passivation is a chemical process that generally only affects metals and involves the production of a film by reaction with the surrounding environment. This thin layer insulates the stent and makes it neutral to the organism. In addition to choosing the right type of stent, the size of the stent is also very important. In determining the diameter of stents, a scale defined by the French surgical instrument maker Joseph-Frédéric-Benoît Charrière is commonly used [22]. The abbreviation CH, or Ch, from his last name, was used to refer to the unit of measurement, but nowadays it is more common to use the abbreviation from the French word ‘Fr’ or ‘FR’. The abbreviations ‘Fg’ or ‘F’ may also be encountered, although these are rarely used. This unit defines the outer diameter of the catheter. To convert the diameter of the catheter into millimetres, the French size is divided by 3 (1 mm = 3 Fr).

For example, typical ureteral stent sizes for cats are 2.5 Fr, while for dogs they are 3.7, 4.7, and 6 Fr [23]. Since 2000, the interest in stents in veterinary medicine has grown considerably due to many publications and retrospective reports on their functionality, usefulness, side effects, and complications. In summary, this review will discuss the current state of knowledge on the use of stents in veterinary medicine.

## 2. Types of Stents Used in Veterinary Medicine

The selection of proper material in relation to the stent placement site has enormous significance. The creation of the ideal stent has been a priority in interventional surgery for more than 30 years, but it is still difficult to combine all the characteristics that meet this criterion: elasticity, biocompatibility, thermostability, resistance to pressure forces, and a lack of reaction to surrounding tissue. Its properties determine the subsequent possibility of postoperative complications, which sometimes necessitate a second procedure. Currently, the most commonly used metal material in veterinary medicine is nitinol. It has gained its popularity due to two important characteristics, super-elasticity and shape memory. The super-elasticity of nitinol stents is defined by their incredible ability to adapt their shape in relation to the high mobility of a given structure, such as a vessel or trachea [24]. However, in some cases, defects related to stent flexibility are still reported, as in the case of peripheral artery disease in humans during stenting of the femoro-popliteal arteries [25]. Although in animals there is no predilection for atherosclerotic plaque accumulation in vessels, the site associated with high mobility is the trachea during rapid head movements, especially in impulsive dogs [26]. Shape memory, on the other hand, prevents any deformation of the stent due to the high temperature > 20 °C, thus disallowing any shape change after insertion due to the constant body temperature of the animals [24]. Another equally important property is its biocompatibility with surrounding tissues, which prevents inflammatory reactions and responses [24]. Other metals are not as common as nitinol, however, the use of materials such as stainless steel or elgiloy, which has very similar properties, has been reported [27,28]. 

Polymeric fibres are a forward-looking material used in human medicine and, more importantly, in veterinary medicine. In animals, they are used for urinary tract stenting with double-pigtail stents or for stenting the blood vessel wall with drug-eluting stents, which can be temporary—biodegradable—or permanent. Polymer stent materials include polyurethane, silicone, poly-lactide-co-glycolide, or l-lactide-glycolic acid copolymer [29]. Maintaining vessel patency in animals is rarely performed because vessel wall obstruction in animals generally occurs when other structures such as cancerous tumours are compressed [30]. In humans, the problem is much more exacerbated due to a predilection for the accumulation of atherosclerotic plaques concentrating on the inner vessel wall, leading to stenosis of the vessel lumen [31]. Stenting of the urinary tract is observed much more frequently in everyday veterinary practice and polymeric materials are the main raw material for ureteral stents. Postoperative complications related to restenosis, stent migration, or tissue encrustation are still a major problem. For this reason, developers are looking for newer and better solutions, with an emphasis on greater biocompatibility, durability, and flexibility. Future solutions using biodegradable polymers or metal drug-eluting stents for specific conditions are being considered [32]. However, the use of drug-eluting stents in veterinary medicine is poorly understood and requires much research into their efficacy and performance in animals, as evidenced by the small amount of documented work in this area. Materials for the production of stents in veterinary medicine are listed in Table 1. The data presented suggest that nitinol is the most commonly used material, mainly used for stenting of the respiratory tract, urethra, and some vessels, but in the latter case, it is much less common. This choice is undoubtedly due to the properties of nitinol described above, as well as the many papers available in the literature on efficacy and post-operative complications, which only confirms that nitinol may be the principal material for stents. The choice of this material in the context of ureteral stenting is not so obvious anymore, as the most commonly used materials in this type of surgery are polymeric fibres, mainly polyethylene. However, there is still too little information available on the efficacy of biodegradable and drug-eluting stents in the context of successful therapy and the impact on the animal body, which should certainly become a topic for future research. 

## 3. Stents in Airway Collapse

The most common stenting procedures performed in veterinary medicine are aimed at supporting the structure of the tracheal wall or trachea with the initial sections of the bronchial tree. Indications for stent placement are drug-resistant collapses of the respiratory tract, termed: tracheomalacia, tracheobronchomalacia, or bronchomalacia. Breeds predisposed to tracheomalacia and tracheobronchomalacia are small breeds such as Yorkshire Terriers, Chihuahuas, Pomeranians, Mops, and the Miniature Poodles [33,54]. Cases of primary bronchial collapse are also encountered in large-breed dogs [54]. Symptoms occur most commonly in middle-aged dogs, but severe cases in younger and older individuals have also been reported [55]. The aetiology is multifactorial and not fully understood. For this reason, it is classified as both a primary (genetically determined) and secondary (dependent on environmental factors) condition. Among the secondary factors, microorganisms, microclimate, macroclimate, and co-morbidities, e.g., cancer, are most commonly mentioned [56]. Histopathological studies have shown that airway collapse is accompanied by a loss of significant amounts of glycosaminoglycans, chondroitin sulphate, and a significant decrease in water-binding capacity, leading to pathological changes in the structure of the cartilage tissue [57]. Tracheal collapse and stent placement are shown in Figure 3. The first attempts at non-pharmacological treatment concentrated on supporting the tracheal cartilage using extraluminal polyurethane rings. However, these treatments often resulted in serious complications, such as paralysis of the recurrent laryngeal nerve [58] or tracheal damage [59]. Furthermore, short- and long-term analyses indicated high case mortality [34]. Additionally, the procedure was associated with high invasiveness, as a considerable neck dissection was required to access the trachea. Furthermore, if the airway collapse involves the thoracic segment, the procedure is impossible to perform due to the presence of bony scaffolding (sternum and ribs), preventing atraumatic access to the trachea [34]. For this reason, an intraluminal technique using stents has become the first-line procedure. In a retrospective study of 75 dogs with the tracheal collapse in which a nitinol stent was placed, it was shown that this procedure had satisfactory results and was associated with long survival of the operated animals. In most cases, symptoms associated with the disease, i.e., the characteristic cough described as having a goose-hong or raspy sound, disappeared after the procedure [60]. These observations were later confirmed by other authors [26,35,36]. However, despite being much less invasive, airway stenting also carries a risk of complications. The complications that have been reported by researchers have mainly concerned the insufficient strength of the stents, which migrated in very active dogs, and became deformed (shortened) or broke [28,34,36,60]. Less common complications after stent placement were airway mucosal hyperplasia with ingrowth into the stent structure, and bacterial and non-bacterial airway inflammation [34,36,60]. Perineal hernia and rectal prolapse were also noted in one of the mentioned studies [36]. Secondary collapse of the tracheal or bronchial wall near the proximal and distal ends of the stent has also been reported [61]. For this reason, supporting the trachea with a stent along its entire length is recommended to prevent similar situations. A reduction in the frequency of disorders associated with mechanical failure of the stent became possible after the introduction of the nitinol ‘Fauna’ stent, which is a combined cross-hook braided stent—previous stents were cross-braided [37]. This modification made it possible for the stent to adapt to the torsional and compressive forces resulting from the animal’s movement, preventing migration or fracture of the stent. Results on 22 dogs confirmed that postoperative complications were lower: those related to stent migration were only 4.5%, while fractures occurred in 9.1% [39]. In earlier studies, this was 19% for migration [60] and 25% for fractures [28]. Furthermore, this stent did not exert excessive pressure on mucosal cells due to reduced inflammatory complications [39]. Recently, an innovative way to implant a stent has been proposed. The stent was threaded into the tracheal lumen through a small tracheal incision made on the ventral side of the neck [38]. The innovative shape and method of fixation effectively prevented migration, but also allowed easy removal (twisting out) of the stent, even after full epithelialisation of the nitinol threads. All complications associated with the surgical procedure resolved within one month of the procedure without the need for pharmacological intervention [38]. These results seem promising; however, this study describes effects obtained in only four dogs and does not include long-term observations. Therefore, further studies are needed to fully assess the efficacy of spiral nitinol stents. It is also worth mentioning that the selection of an appropriate stent diameter is of crucial importance in terms of postoperative complications. An inadequate stent size promotes stent migration and fracture [4]. The best method to assess the size of the tracheal lumen seems to be CT scans, but this method is often too expensive for pet owners [62]. Therefore, most measurements are based on radiographs from the dorso-abdominal and/or lateral position [63]. Usually, treatment starts with pharmacotherapy and the use of stents is only considered when this proves to be ineffective. Intraluminal airway maintenance is the only effective management option and is likely to become the first-line treatment in the future. This is supported by excellent results, expressed not only in a significant improvement in the comfort of life but also in an increase in the duration of life. 

Cases of the use of stents in the treatment of nasopharyngeal stenosis have also been described. Despite the risk of serious complications, e.g., palatal defect, this treatment results in a significant improvement in the comfort of the animal [64]. 

## 4. Stents in Urinary Tract Obstructions 

Urinary tract obstructions are the second most commonly described use of stents in veterinary medicine. This applies to both the ureters and the urethra. It appears that ureteral stenting is more common, however there are an increasing number of reports documenting urethral patency with stents.

## 5. Ureter

Ureteral stenting involves removing the obstruction of the ureter lumen and obtaining a renewed ability to drain urine from the kidney into the bladder. The most common indication is urinary tract stones. In this case, the use of a stent allows passive dilatation of the ureter and drainage of stones, although this is not always the procedure performed first [65]. A study of 5230 feline specimens showed different compositions of the minerals from which these stones can be formed, but the most commonly diagnosed are calcium oxalate and ammonium magnesium phosphate (struvite) [66]. These findings have also been confirmed in dogs [67]. Breed predisposition, age, housing, or overall serum calcium levels are not important in the aetiology of this condition in cats, but the intake of dry food-only diets significantly increased the likelihood of its occurrence [68]. In dogs, small canine breeds such as the Yorkshire Terrier, Miniature Schnauzer, and Miniature Poodle are most commonly affected by calcium oxalate stones [69], and castrated males are more often affected [70]. Noteworthy is the Dalmatian breed, in which the genetic predisposition to uric acid stone formation is due to a disorder of purine and pyrimidine metabolism [71]. The diagnosis of ureteral obstruction can be made by abdominal ultrasound, non-contrast radiography, but even better by contrast radiography (ureteropielography) in the lateral or dorsoventral position in most cases [72]. Other indications for the insertion of a ureteral stent include: stricture of the ureter due to its inflammation, damage during surgery (ureteral rupture), and strictures caused by cancer, e.g., transitional cell carcinoma localized in the bladder triangle [47,48,73]. Furthermore, conditions occurring within the kidney, such as pyelonephritis and hydronephrosis, can lead to blockage [6,50], and in such cases, veterinarians often decide to place a double-pigtail stent [53]. Its name derives from the way the stent is constructed because both ends of the stent are rolled up in the shape of a pigtail, which allows it to be fixed in both the urinary bladder and the renal pelvis. A radiograph illustrating a double-pigtail stent is shown in Figure 4. The technique itself is quite complex and requires experience as well as a special apparatus. The stent is placed anterograde, i.e., the ureter is stented from the kidney side by subcutaneous implantation under fluoroscopy guidance or by the surgical opening of the abdominal cavity and insertion of the stent under visual control. The second procedure is a post-grade technique involving the insertion of a stent into the ureter from the urethral side [52]. This procedure is considerably more difficult in cats due to the smaller diameter of the ureter. In addition to the benefits of the procedure, it is also important to mention the variety of complications resulting from ureteral stenting, which can be divided into mild and severe. Some authors in their studies presented complications occurring during the operation and in the follow-up of the effects of ureteral stenting in the short, medium, and long term [51]. While complications up to one month were not significant and occurred in low numbers, complications in the follow-up period were more frequent and mainly included infections [74]. Urinary tract infections were most commonly caused by *Enterococcus* spp. [74]. Many were subclinical, but some caused significant clinical symptoms, i.e., dysuria, ureteral restenosis, or haematuria. The most commonly described complications from stenting of the different systems are shown in Table 2. A safer technique (especially in cats) seems to be the subcutaneous ureteral bypass (SUB) at present. This is a method that allows urine to flow from the renal pelvis to the bladder, bypassing the ureters through a subcutaneously implanted shunting port [51,75]. This is an expensive technique, which limits its use in clinical practice, and therefore greater hopes are placed on the improvement of stents.

## 6. Urethra

Urethral stenting is a less frequently performed procedure compared to ureteral stenting, but an increasing number of cases of urethral obstruction are being reported. The diagnosis of urethral obstruction is mainly based on radiography with the use of contrast [5]. Indications for stent placement include malignant as well as non-malignant causes of urethral obstruction. Non-malignant causes include chronic mucosal oedema, functional idiopathic urethral obstruction, or interruption of the urethral wall [5,49]. The malignant category includes cancers that are not necessarily directly related to the urethral structure, but to their size, which leads to urethral compression and closure. These are most commonly transitional cell carcinoma, prostate cancer, or hemangiosarcoma [41,76]. In such cases, stenting is a form of palliative treatment to relieve the animal’s discomfort and enable urination. This procedure very often provides the animal with great relief, but the positive effects may be short term, as further tumour growth may lead to re-closure of the urethral lumen [76]. Other complications include mild to severe incontinence, stranguria, secondary urinary tract infections, stent migration, or stent fracture. A single case of urethral wall puncture by a guidewire due to an error by the operating physician has been described [40,41,43]. Stents implanted in the area of the proximal urethral sphincter, which is responsible for will-dependent urination, can cause severe incontinence [42]. This appears to be supported by another study, where the only dog in the entire study group with a temporary stent placed in the urethra outside the area of its proximal sphincter, had will-dependent urination, while the rest of the individuals with stents placed in the proximal section involving the sphincter had incontinence [5]. Urethral stenting in the form of self-expandable nitinol stents is often too expensive for the owner, so the authors suggest that the previously mentioned temporary stents made from materials available in almost any clinic may be an alternative. They require a catheter with two drainage holes to serve as a stent-like duct and a needle along with surgical thread to be passed through its structure to produce a loop at the end, which will discharge into the vaginal vestibule. The loop is to serve as an anchor for the surgical suture proper, which will be sewn through the uterine wall and tied to the skin. While this is associated with a high probability of incontinence, it is of great benefit in the context of healing of the ruptured epithelium, where, after epithelialisation, the temporary stent can be easily removed [5]. Stenting of the urinary tract in case of urinary tract strictures of various aetiologies can significantly prolong the survival time; however, it is still necessary to determine the negative effects that may develop after the procedure, especially in long-term follow-up. For this reason, the animal requires continuous monitoring to avoid or rapidly diagnose relapse or the appearance of severe complications, thereby increasing the chances of survival.

## 7. Stents in Heart Disease

Stents are needed in a few special cases in cardiology where drug treatment or balloon plasty are ineffective, or if the current condition requires the use of a different treatment strategy. One of the most common congenital heart defects in dogs is pulmonary artery stenosis [77]. It is most commonly caused by deformation or fusion of the pulmonary artery valve leaflets and hypoplasia of the pulmonary artery fibrous ring. The development of the disease often exacerbates the clinical symptoms, causing exercise intolerance, right-sided heart failure, and a general deterioration of the health associated with abnormal blood distribution in the body [78]. The serious limitations of performing such a procedure include its cost, as well as the small number of veterinarians capable of performing it. Nonetheless, long-term analyses after pulmonary artery stenting procedures are promising [79]; therefore, this technique should be considered as one of the alternative treatment options compared to the classic methods, such as balloon valvuloplasty.

Another equally rare indication for stenting is the widening of the opening in the pathological membrane dividing the right atrium—a persistent venous valve in a defect called cor triatriatum dexter. Membrane stenting is an alternative treatment to balloon plasty, which is standard in such cases, albeit is infrequently performed [80,81]. 

Stenting of the caudal vena cava has also been successfully applied in a dog with chronic stenosis of the pathological junction of the azygos vein and the caudal vena cava. The defect leads to renal thrombosis and secondary chronic renal failure. After stent placement, the clinical signs resolved, and the abnormal veno-venous anastomosis did not adversely affect the dog’s continued life [82]. Stenting of the second major vein was reported in the setting of a tumour infiltrating the heart wall, causing compression of the intracranial vena cava and increased blood pressure. Although the stenting was only a form of palliative treatment, the clinical symptoms of exercise intolerance and lymphatic accumulation (chylothorax) resolved, and the condition significantly improved for another six months [31].

## 8. Stents in the Treatment of Liver Disease

Stents can be used to support the walls of the hepatic veins and part of the caudal vena cava in Budd–Chiari syndrome—a rare developmental disease associated with impaired venous blood outflow from the liver. Performing this procedure yielded highly encouraging results. Animals undergoing stenting showed a significant improvement in clinical condition, and the survival time was extended from 7 to 20 months [44,45].

The use of stents also works well for the treatment of chronic bile duct obstruction [46].

## 9. Stents in the Treatment of Gastrointestinal Diseases

Stents have been used to restore gastrointestinal function, e.g., obstruction of selected sections of the large intestine. This occurs as a result of cancerous growths that compress the intestinal walls, causing the closure of the lumen. Stenting is a form of palliative treatment to reduce symptoms, relieve pain, and allow faecal passage [8].

## 10. Stents in the Treatment of Reproductive Tract Disease

Literature data suggest that a cervical stent can be used during ongoing pyometra to drain accumulated pus in the uterus. Cervical stenting has yielded very good results without the occurrence of serious complications [83].

## 11. Limitations

All studies cited in the text were retrospective in character. Cases were recruited after the owners reported having noticed signs of disease. Consequently, none of the studies described the clinical condition before the onset of clinical signs, including the full health of the animal, and therefore, it is not possible to unequivocally determine whether the animals completely recovered from the obstruction of the structure in question. Consequently, this does not give a complete picture of the efficacy of the method in question, but it seems impossible to carry out such studies under laboratory conditions.

## 12. Future Perspectives

Stenting in veterinary medicine is a highly successful alternative treatment for stenosis in some body systems. However, despite a high success rate, there are still cases of post-operative complications that make life more or less difficult for the specific case. In the case of maintenance of airway obstruction, a solution involving greater strength and elasticity of the stent itself should be sought in order to prevent fractures. In addition, migration of these stents is also observed, so solutions aiming at a more permanent bond with the inner tracheal wall are also advisable, as in the case of the spiral tracheal stent. A higher incidence of complications is reported after placement of ureteral stents, which emphasises the need for biodegradable ureteral stents in the future, allowing for reduced pain, stranguria, or incontinence. Metal drug-eluting stents are also desirable due to the reduction of excessive encrustation of the ureteral epithelium or the amount of bacterial inflammation. However, in the context of drug-eluting stents, further studies are needed to assess their efficacy and effects in the animals. 

## 13. Conclusions

As the field of veterinary surgery develops, new alternatives to classic treatment methods are being found. Stents are therefore increasingly used in veterinary medicine and are becoming a valuable addition to standard treatment strategies for canine and feline diseases. Despite the relatively high cost of treatment, their use makes it possible to improve comfort and prolong life. In many cases, it is a life-saving treatment. The improvement of stents, in terms of materials, their design, and the continuous improvement of the technique of their insertion and stabilization at the target site, will certainly lead to an increase in the popularity of this treatment technique. 

## Figures and Tables

**Figure 1 materials-16-01480-f001:**
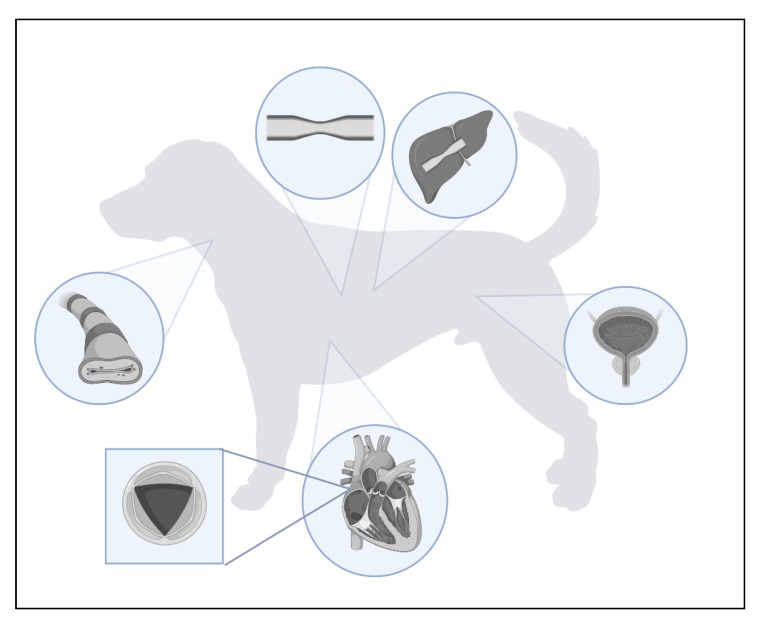
Use of stents in the treatment of canine and feline diseases.

**Figure 2 materials-16-01480-f002:**
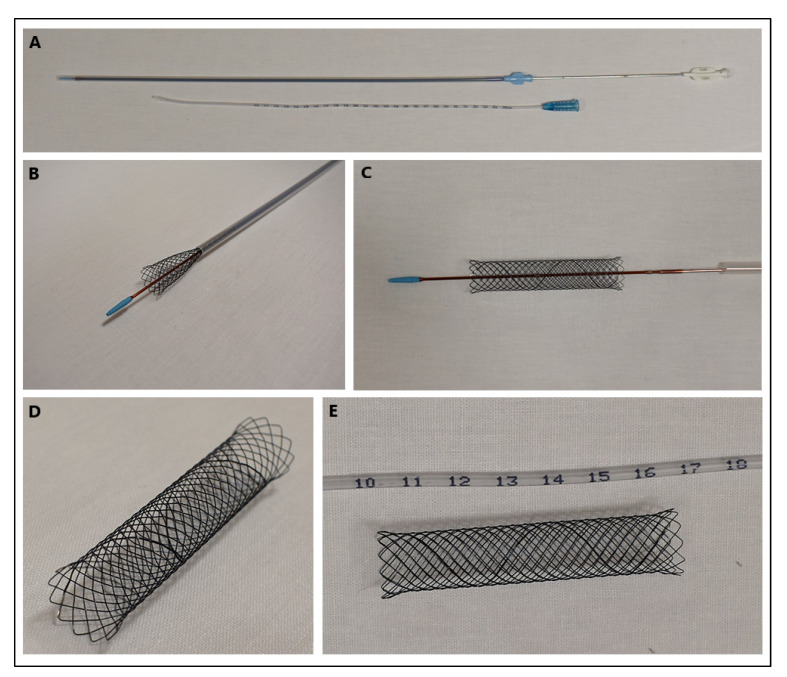
(**A**) Tracheal stenting kit containing a nitinol stent with a cross-braid and a measure to assess the tracheal length. (**B**) View of the initial stent deployment process. (**C**–**E**) Different views of the fully expanded stent. The photographs come from the private archives of Dr Dariusz Jagielski.

**Figure 3 materials-16-01480-f003:**
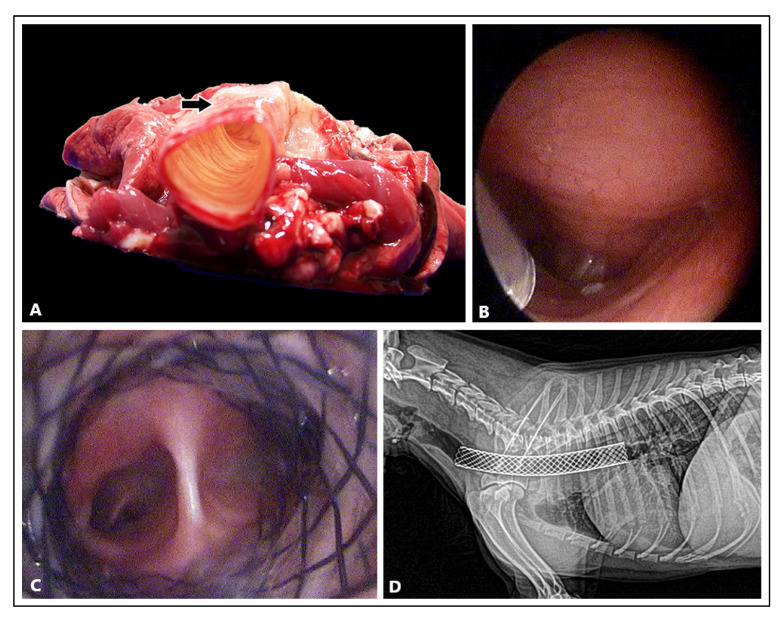
(**A**) Autopsy of a 15-year-old male dachshund euthanized due to end-stage cardio-respiratory disorders related to chronic disease of the mitral valve and second degree tracheal collapse. The tracheal rings are deformed and the tracheal muscle closing the dorsal ring is overstretched and flaccid. The collapse of the muscle limited the lumen of the trachea on expiration and caused shortness of breath. (**B**) Endoscopic examination of an 8-year-old male Yorkshire Terrier with third degree tracheal collapse. (**C**) Endoscopic examination after endotracheal stent implantation. (**D**) X-ray of the chest and neck in the lateral view after insertion of the stent. The photos were taken during the veterinary services of Dr Robert Paslawski.

**Figure 4 materials-16-01480-f004:**
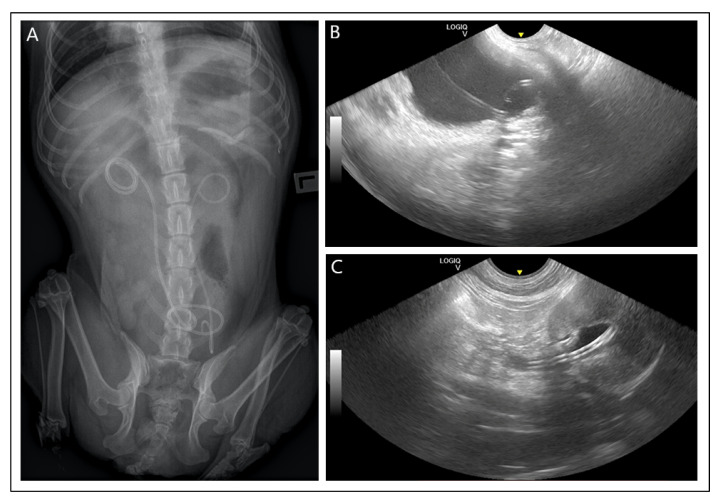
(**A**) Dorsoventral X-ray view of an animal with a double-pigtail stent positioned bilaterally in the ureters. (**B**) Ultrasound image of the end of a pigtail stent in the urinary bladder. (**C**) Ultrasound image of a pigtail stent placed in the renal pelvis. The photos were taken during the veterinary services of Dr Dariusz Jagielski.

**Table 1 materials-16-01480-t001:** Materials and composites used in the fabrication of stents, including the site of insertion in different animal species.

Type of Stent	Stent Material	Stent Placement Site	Animal	Author
Metallic	Nitinol	Trachea	Dog	[33] [34] [35] [36] [26] [37] [28] [38] [39]
		Urethra	Dog	[40] [41] [42] [43]
		Colon	Dog	[8]
		Azygos vein	Dog	[44]
		Hepatic vein	Dog	[45]
		Caudal Vena Cava	Dog	[46]
	Elgiloy	Trachea	Dog	[28]
Biodegradable	Polylactide	Bile ducts	Dog	[13]
	Polydioxanone	Bile ducts	Porcine	[14]
	Polycaprolactone	Trachea	Rabbit	[15]
Polymeric	Polyethylene	Ureter	Cat	[47] [48] [49] [50]
		Ureter	Dog	[51] [52] [6] [53]
Silicone	Medical-grade silicone	Larynx	Dog	[16]
Drug eluting	Paclitaxel	Coronary artery Iliac artery	Porcine Rabbit	[12]
		Bile ducts	Dog	[13]
		Bile ducts	Porcine	[14]
	Zotarolimus	Coronary artery Iliac artery	Porcine Rabbit	[12]
	Biolimus/sirolimus	Coronary artery	Porcine	[11]
		Iliac artery	Rabbit	[12]
		Femoral artery	Dog	[17]
	Cefotaxime	Bile ducts	Dog	[19]
	Arsenic trioxide	Iliac artery	Rabbit	[10]
	Triclosan	Ureter	Rabbit	[18]

**Table 2 materials-16-01480-t002:** Some complications after the stent placement procedure.

Stent Placement Site	Complications	Authors
Trachea	Stent migration	[34] [39]
	Stent fracture	[36] [33] [35] [34] [39] [28] [60]
	Re-obstruction	[34] [39] [28] [60]
	Pneumonia	[36] [61] [35]
	Excessive granulation tissue	[35] [28] [60]
	Tracheitis	[35] [34]
	Coughing	[61] [34]
Ureter	Re-obstruction	[74] [47] [48] [50] [52]
	Excessive granulation tissue	[48] [6]
	Migration	[74] [48] [6]
	Ureteritis	[74] [48]
	Dysuria	[74] [48]
	Pyelonephritis	[74] [48] [75]
	Reflux	[48]
	Sterile cystitis	[49]
	Penetration of ureter wall	[52]
	Haematuria	[74] [52]
Urethra	Incontinence	[42] [41] [40]
	Stranguria	[42]
	Re-obstruction	[41] [40]
	Migration	[41]
	Penetration of urethral wall	[43]

## Data Availability

Not applicable.

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
