# Peer review of "Stents in Veterinary Medicine"

_materials, 2023, doi:10.3390/ma16041480_

Round 1
Reviewer 1 Report
1. Extensive editing of English language and style required.The first sentence of the abstract needs to be reorganized.
Stenting in veterinary medicine has been a rapidly growing method of interventional surgery for respiratory, urinary, vascular and other tract collapses for several years.
2.it seems impossible to carry out such studies under laboratory conditions.why?
Author Response
Manuscript ID materials-2046932
Dear Academic-Editors and Reviewers,
Thank you for inviting us to respond to the very thoughtful and constructive reviewer comments. We greatly appreciate the reviewers time and believe our revised manuscript has become more well-rounded as a result.
We have incorporated all suggestions throughout the manuscript, and changes are highlighted in red.
Below is a point-by-point response to reviewers’ and Editor’s comments to clarify which edits were made.
We are happy to respond to additional requests if they arise.
Sincerely,
Szymon Graczyk

Reviewer 2 Report
In the present article, the authors threw the light on the uses of different stents in the veterinary practice. The article was well organized, and presented. However, the following are some points to consider.
1. "Their use in medicine began in 1986 when the first self-expandable stent was implanted in the coronary arteries with long-term positive re-sults [1]."
Since the present article discuss the stents in the veterinary practice, it is advised to choose a reference related to the veterinary medicine.
2."Since 2000, the interest in stents in veterinary medicine has grown con-siderably through which many papers and retrospective reports have been written on their functionality, usefulness, side effects and complications, so in summary, this review will discuss the current state of knowledge on the use of stents in veterinary medicine."
This sentence is very long. it will be easier to understand if separated into two shorter sentences.
3. Please insert the copyrights of the figures 2, 3, and 4 and mention in the legends of these figures.
4.In table 1, please discuss the outcomes of the use of the stents described in the table.
5. In tables 1, and 2, the refence style is different than that in the text, please unify the refence style.
6."Limitations
All studies cited in the text were retrospective in character. The group of patients was collected over time after the owners reported having noticed signs of disease."
It is better to use the word cases instead of patients as the focus of the article is the veterinary medicine. please unify among the entire of the manuscript.
Author Response

(The authors gave the same response as above.)

Reviewer 3 Report
The paper examines Stents in Veterinary Medicine. The topic is very relevant since its representing a rapidly growing method of interventional surgery for respiratory, urinary, vascular, and other tracts. It consists of maintaining the permeability of a given tubular structure, thus allowing the passage of gas or liquid. The paper presents novel and useful findings. The introduction provides evidence-based background for the research. All sections have been properly described, all gathered results are well presented and data interpretation is appropriate. The findings of all authors are thoroughly discussed, and limitations, recommendations, and conclusions are justified. I did not find any objective errors, and my suggestion is to accept the manuscript in its current form.
Author Response

(The authors gave the same response as above.)

Reviewer 4 Report
The aim of this review was described the several uses of stenting as a method of interventional surgery in veterinary medicine and they also highlight this procedure as an alternative method, even cheaper than others, which consists of maintaining the permeability of a given tubular structure, being a lifesaving or comfort improving procedure for animals in different pathologies as tracheal collapse, colonic obstruction, and others. However, it is also important to know the other side, I mean, the postoperative complications associated with these procedures, that the authors have managed to focus well in this review.
For this reason, I consider that it should be accepted in its present form.
Author Response

(The authors gave the same response as above.)

Reviewer 5 Report
Very interesting review.
Author Response
Detailed answer to reviewer 1
Reviewer 1: Very interesting review.
Answer: We thank the reviewer for his/her comment, which we appreciate.

Reviewer 6 Report
The manuscript is very extensive Iwould say too much. This is a review manuscript not a treatid. The tables should be clearer. Characteristics should not be done with author citations
There are many references but some very importan ones are missing
Author Response
Detailed answers to reviewer 2
Reviewer 2: The manuscript is very extensive I would say too much. This is a review manuscript not a treatid.
Answer: We thank the reviewer for his/her erudite comment, which we appreciate. So far, the literature has been missing a summary that brings together the knowledge of stenting of different systems in the animal organism. Admittedly, there have been papers presenting data on stenting of the respiratory or urinary tract, but these have been presented separately. Therefore, this review, somewhat lengthy, in our opinion fairly presents information in the field of stenting in veterinary medicine.
Reviewer 2: The tables should be clearer. Characteristics should not be done with author citations
Answer: In the case of Table 1, we wanted to provide a clear representation of the site of stent application along with the most commonly used stent material, so it was constructed as clearly as possible. As for Table 2, our goal was to present possible complications after stent placement in the most commonly stented body systems, i.e. the respiratory and urinary tracts, so we tried to shorten the number of columns to the smallest number to make it easier to read and understand.
Reviewer 2: There are many references but some very important ones are missing.
Answer: We have tried to ensure that the citations used fully capture the essence of the problem, so despite their extensive number, they have been selected for the purpose of this article.
Round 2
Reviewer 1 Report
The first draft of the paper can be published after revision.
Author Response
Detailed answer to reviewer 1
Reviewer 1: The first draft of the paper can be published after revision.
Answer: We thank the reviewer for his/her comment, which we appreciate.
